# Modelling the health impact of food taxes and subsidies with price elasticities: The case for additional scaling of food consumption using the total food expenditure elasticity

**Tony Blakely**[1,2]*, **Nhung Nghiem**[2], **Murat Genc**[3], **Anja Mizdrak**[2], **Linda Cobiac**[4], **Cliona Ni Mhurchu**[5], **Boyd Swinburn**[6], **Peter Scarborough**[4], **Christine Cleghorn**[2]

1 Population Interventions Unit, Melbourne School of Population and Global Health, University of Melbourne, Melbourne, Australia, 2 Burden of Disease Epidemiology, Equity and Cost Effectiveness Programme, Department of Public Health, University of Otago, Wellington, New Zealand, 3 Department of Economics, University of Otago, Dunedin, New Zealand, 4 Nuffield Department of Population Health, Oxford University, Oxford, United Kingdom, 5 National Institute of Health Innovation, University of Auckland, Auckland, New Zealand, 6 School of Population Health, University of Auckland, Auckland, New Zealand

* ablakely@unimelb.edu.au

**Data Availability Statement:** There are multiple sources of data used in this study, to parameterize the diet and multistate lifetable models. The sex,

## Abstract

### Background

Food taxes and subsidies are one intervention to address poor diets. Price elasticity (PE) matrices are commonly used to model the change in food purchasing. Usually a PE matrix is generated in one setting then applied to another setting with differing starting consumptions and prices of foods. This violates econometric assumptions resulting in likely mis-estimation of total food consumption. In this paper we demonstrate this problem, canvass possible options for rescaling all consumption after applying a PE matrix, and illustrate the use of a total food expenditure elasticity (TFEe; the expenditure elasticity for all food combined given the policy-induced change in the total price of food). We use case studies of: NZ$2 per 100g saturated fat (SAFA) tax, NZ$0.4 per 100g sugar tax, and a 20% fruit and vegetable (F&V) subsidy.

### Methods

We estimated changes in food purchasing using a NZ PE matrix applied conventionally, and then with TFEe adjustment. Impacts were quantified for pre- to post-policy changes in total food expenditure and health adjusted life years (HALYs) for the total NZ population alive in 2011 over the rest of their lifetime using a multistate lifetable model.

### Results

Two NZ studies gave TFEe's of 0.68 and 0.83, with international estimates ranging from 0.46 to 0.90 (except a UK outlier of 0.04). Without TFEe adjustment, total food expenditure decreased with the tax policies and increased with the F&V subsidy–implausible directions of shift given economic theory and the external TFEe estimates. After TFEe adjustment, HALY gains reduced by a third to a half for the two taxes and reversed from an apparent health loss to a health gain for the F&V subsidy. With TFEe adjustment, HALY gains (in

age and ethnic composition of the NZ population in 2011 was set using Statistics NZ resident population estimates. All-cause mortality and morbidity data, and disease specific incidence, case fatality and prevalence rates, are all provided in an Excel file "Disease Inputs used for Multi-State Life Table Modelling (Version 1.0)" available at the BODE3 publications webpage: https://www.otago.ac.nz/wellington/departments/publichealth/research/bode3/publications/index.html. Note that many of these data are generated specifically for the DIET model, requiring 'epidemiological coherence' using the DISMOD epidemiological calculator.1 They may not be fit for other purposes. Excess health system costs for each disease included in the model are provided in the DIET model Technical Report 2 - also available at the above BODE3 publications webpage. The price elasticity matrix is reported in S1 and S2 Tables of the current paper. Relative risks for the risk factor disease associations are in the DIET model Technical Report.2 Starting food consumption at the level of 340 foods, by sex and ethnicity, was taken from the NZ Adult Nutrition Survey; as stated in the paper, was provided by the Otago University 'Life in New Zealand' (LINZ) staff (personal communication, Blakey, Smith and Parnell, 2014). Our data access agreement does not allow full disclosure of the estimates we extracted from this data, but others can approach LINZ directly for the data. Starting food prices were sources from Nutritrack data, which is available upon request from the National Institute of Health Innovation, University of Auckland. Other key variables, such as the total food expenditure elasticity (TFEe), are detailed in the manuscript. 1. Barendregt J, Oortmarssen GJ, Vos T, Murray CJL. A generic model for the assessment of disease epidemiology: the computational basis of DisMod II. Popul Health Metr 2003;1(1):4. 2. Cleghorn CL, Blakely T, Nghiem N, Mizdrak A, Wilson N. Technical Report for BODE3 Diet Intervention and Multistate Lifetable Models, Version 1.1. Burden of Disease Epidemiology, Equity and Cost-Effectiveness Programme - Technical Report No16. Wellington: Department of Public Health, University of Otago, Wellington 2018.

**Funding:** Health Research Council of New Zealand Programme Grants: Burden of Disease Epidemiology, Equity and Cost Effectiveness Programme (10/248); Effective interventions and policies to improve population nutrition and health (13/724). The funders had no role in study design, data collection and analysis, decision to publish, or preparation of the manuscript.

1000's) were: 1,805 (95% uncertainty interval 1,337 to 2,340) for the SAFA tax; 1,671 (1,220 to 2,269) for the sugar tax; and 953 (453 to 1,308) for the F&V subsidy.

## Conclusions

If PE matrices are applied in settings beyond where they were derived, additional scaling is likely required. We suggest that the TFEe is a useful scalar, but we also encourage other researchers to examine this issue and propose alternative options.

## Introduction

Nutrition policy to prevent or mitigate the obesity epidemic, and improve diets, is a major policy issue.[1] One policy option is food taxes and subsidies.[2, 3] To guide policy making, an important role of research is to estimate the likely impact of such taxes and subsidies on changes in diet (e.g. consumption of fruit and vegetables, total energy intake) [4, 5], intermediate outcomes (e.g. body mass index (BMI) [6], blood pressure), disease outcomes (e.g. stroke, diabetes) [7] and 'total' health measures change (e.g. deaths averted, disability adjusted life years averted or quality adjusted life years gained).[8, 9] Ideally, one would have randomized trials of food taxes and subsidies for this estimation (e.g. [10–13]), but they are difficult to implement and if implemented follow up for long-periods of time to ascertain both long-run accommodation to new food prices and health outcomes is often not feasible. Alternatively, one can analyze natural experiments, e.g. the tax on sugar-sweetened beverage (SSB) in Mexico [14, 15], the Danish saturated fat tax [16] and the SSB tax in Philadelphia.[17]

To estimate the health impacts, including the relative health impacts across multiple policy options, modelling is therefore useful.[18] A key–and challenging–aspect of this modelling is parameterizing how total diets actually change with food taxes and subsidies. For example, how much does a tax on one food item affect consumption of other foods? Price elasticities (PEs) are commonly used to convert a food tax/subsidy intervention to a change in total diet [19] which is then linked to changes in BMI and other risk factors, then disease rates, and then to morbidity and mortality.

There are two types of PEs. First, there is the own-PE. This measures how much the price (change) of a given food affects its own consumption. For example, a PE of -0.7 on red meat means that if the price of meat increases by 1%, its consumption reduces by 0.7%. The second is cross-PEs. This measures how much the price (change) of another food affects consumption. For example, a cross-PE of +0.1 for red meat given an increase in price of poultry means that if the price of poultry increases by 1%, the consumption of red meat increases by 0.1%. (A positive cross-PE means foods are substitutes and a negative cross-PE that they are complements)

Econometric analyses to generate PEs are demanding, requiring simultaneous data on both price and demand and (ideally) substantial variation in price (e.g. by time or by region). Given the impracticality of calculating new PEs for each new setting in which researchers and analysts estimate the impact of price changes on food purchasing, by necessity PEs from one setting (e.g. a given country for a given year) are often applied in another setting.[5, 6, 8] But there are inevitably variations between settings in food purchasing patterns, prices and demand relationships. Indeed, the most common food price elasticity matrix is a 'conditional' one meaning that it is generated under the assumption that there is no change in total food expenditure for food price changes; this nil impact on total food expenditure is violated when a PE matrix is applied to a different distribution of food prices and consumption–likely resulting in implausible shifts in total food expenditure. The Text Box gives a simple food system example, showing how the transfer of price elasticities calculated in one setting of food consumption to another can generate (likely) implausible estimates of post price change consumption.

**Competing interests:** The authors have declared that no competing interests exist.

## Box 1. Problems transporting price elasticity from one context to another

Imagine a world with just three foods: fruit, vegetables and cereal. Average daily per person consumption, price per 100g and kJ of energy per 100g are as follows:

| | kJ per 100g | Price per 100g | own-PE [fruit - fruit] | cross-PE [Fruit-vegetables/cereals] |
|---|---|---|---|---|
| Fruit | 150 | $0.40 | -1 | |
| Vegetables | 150 | $0.50 | | 0.30 |
| Cereals | 500 | $1.00 | | -0.05 |

The own- and cross-PE estimated for this world are also shown. A 1% increase in the price of fruit will result in a: 1% decrease in consumption of fruit; a 0.3% increase in the consumption of vegetables (a substitute food); and a 0.05% decrease in cereals (a complement food in this world). Now imagine that a 20% subsidy on fruit is implemented. The table below shows the pre-subsidy and post-subsidy consumption, energy and expenditure:

| | | g/day | kJ | Expenditure |
|---|---|---|---|---|
| Pre-subsidy | Fruit | 50 (11.1%) | 75 (5.5%) | $0.20 (6.3%) |
| | Vegetables | 200 (44.4%) | 300 (21.8%) | $1.00 (31.3%) |
| | Cereals | 200 (44.4%) | 1,000 (72.7%) | $2.00 (62.5%) |
| | **Total** | **450** | **1375** | **$3.20** |
| Post-20% subsidy on fruit | Fruit | 60 (13.3%) | 90 (6.5%) | $0.24 (7.5%) |
| | Vegetables | 188 (41.8%) | 282 (20.4%) | $0.94 (29.4%) |
| | Cereals | 202 (44.9%) | 1,010 (73.1%) | $2.02 (63.1%) |
| | **Total** | **450** | **1382** | **$3.20** |
| | **Difference pre- to post-subsidy** | **0%** | **0.51%** | **0.00%** |

The subsidy resulted in no change in total expenditure (consistent with the assumptions inherent within the calculation of 'conditional' price elasticities) and increased total energy consumption by 0.51%.

Now imagine we use the above price elasticities in a different setting or population – with differing starting consumption of foods (but the same prices per 100g). The pre- and post-subsidy grams per day consumption, energy intake and expenditure are:

| | | g/day | kJ | Expenditure |
|---|---|---|---|---|
| Pre-subsidy in new setting | Fruit | 100 (25%) | 150 (11.5%) | $0.40 (13.8%) |
| | Vegetables | 100 (25%) | 150 (11.5%) | $0.50 (17.2%) |
| | Cereals | 200 (50%) | 1,000 (76.9%) | $2.00 (69%) |
| | **Total** | **400** | **1300** | **$2.90** |
| Post-20% subsidy on fruit in new setting | Fruit | 120 (28.8%) | 180 (13.5%) | $0.48 (16.2%) |
| | Vegetables | 94 (22.6%) | 141 (10.6%) | $0.47 (15.8%) |
| | Cereals | 202 (48.6%) | 1,010 (75.9%) | $2.02 (68%) |
| | **Total** | **416** | **1331** | **$2.97** |
| | **Difference pre- to post-subsidy** | **4%** | **2.38%** | **2.41%** |

> Note that the total expenditure now changes from pre- to post-subsidy, and the percentage change in total energy consumption (which will largely drive health impacts) is over four times greater than in the original setting (2.38% compared to 0.51%).

This simple example demonstrates distortions that may arise applying price elasticities from one setting to another – in this case just due to differing starting consumptions of foods. There may also be differing starting prices, and more fundamentally differing food preferences with 'genuinely' differing consumer responses to price changes. Ideally, there would be price elasticities worked out for each different setting that they are applied in, however that is impractical.

This price elasticity transferability issue is problematic for health impact modelling studies, which derive health consequences from absolute differences in dietary consumption patterns between the observed and counterfactual scenarios.[18, 20] For example, our intended modelling will be using National Nutrition Survey (NNS) data from 2009 [21] (whilst older, it is nationally representative and has data by ethnicity and for a wide range of food groups), where the price elasticities were generated with experimental data (conducted in 2016, using foods with overlap but not exact concordance with NNS food groups)[22, 23] using Bayesian priors from price elasticities calculated using Household Economic Survey data (2007/08 and 2009/10) and Food Price Index data (2007 and 2010).[5] Is applying PEs derived from one setting into another setting acceptable? We argue yes–but with care (and rescaling). Look again at the Text Box, and in particular the column percentages shown in parentheses in the tables. Whilst we may be concerned about the validity of the new estimated totals of g/day, kJ and financial expenditure post-subsidy, we may accept that the shifts in relative distribution across foods in g/day, kJ and expenditure is valid. Therefore, all estimates need common rescaling given some constraint (preserving the new column percentages). Possible constraints include no change in (or some tolerable change in) one of the total weight of food, the total energy, or the total financial expenditure. The question then is "what constraint should we use?"–a question we consider in more detail below.

The aim of this paper is to peel back often-unacknowledged uncertainties in the use of PEs to model food taxes and subsidies. Our goal is not to 'damn the research endeavour to irrelevance'.[24] Quite the converse. We take the view that improving nutrition is a major public health priority, and it is essential for researchers to estimate the health impacts of food taxes and subsidy options. Modelling is an important part of that research agenda. Accordingly, the objectives of this paper are:

1. Canvass the advantages and disadvantages of options for rescaling total expenditure after conventional application of price changes through a PE matrix, options being: nil change in energy intake; nil change in grams of food; nil change in total expenditure on food; some change in total food expenditure using a total food expenditure elasticity (TFEe; i.e. change in food expenditure is a function of a TFEe and the difference in total food price index (FPI) pre- and post-tax or subsidy policy).

2. Demonstrate the use of the TFEe and its impact (compared to a conventional unscaled application of a PE matrix generated for New Zealand (NZ) [25]) on household expenditure, energy intake, BMI and health adjusted life years gained, for three NZ case studies: a NZ$2 per 100g of saturated fat tax; a NZ$0.4 per 100g of sugar tax; and a 20% fresh fruit and vegetable subsidy.

## Options for rescaling total consumption or expenditure after conventional application of a price elasticity matrix

There are a number of options to rescale all food purchasing after the conventional application of a PE matrix.

1. *Rescale all consumption so that energy intake is unchanged.* Some health impacts of dietary change are mediated through mechanisms other than energy balance and these can plausibly be modelled in the way that Smed et al (2016) did when they estimated the health impacts of the Danish saturated fat tax assuming no change in total energy intake. [16] However, in reality for many pricing interventions, the dietary changes will plausibly change energy balance. While in general humans maintain fairly close energy balance on a week by week basis, small increases or decreases in energy intake maintained over the long term will result in a higher or lower body weight on a year by year basis.

2. *Rescale all consumption so that grams of (solid) food purchased is unchanged.*
Studies by Rolls and others [26, 27] suggest that if one changes the energy density of food, and allow (experimental) subjects to freely eat, then they eat to an amount that keeps the weight of food consumed relatively consistent–and energy intake will thus fall if foods of lower energy density are provided or 'made easier' to access. However, there are limits to an assumption of constant grams of food. First, and an extreme example, a consumer swapping from pre-mixed drinks to powdered sachets will purchase less weight. Second, and less extreme, a consumer swapping from freshly-prepared to dried pasta, or fresh to dried fruit, will purchase less weight. Third, differing moisture contents of substitute foods (e.g. dry cereals compared to moister mueslis) will also presumably not be direct weight substitutes. We are unaware of algorithms to manage these issues if rescaling by food weight was used.

3. *Rescale all expenditure to be unchanged to that pre-tax/subsidy*. This is actually a specific case of the expenditure elasticity approach below. However, it is a simplifying assumption–economic theory suggests that total expenditure on food will change with change in average food price or FPI, as we now explain.

4. *Rescale using total food expenditure elasticity*: Rescale all expenditure using econometric methods that generalize conditional PE matrices (assumed zero change in total expenditure on food due to reducing purchasing for foods with increased prices and/or shifting to other foods; and option 3 above) to unconditional PE matrices (permits change in total food expenditure with food price changes by allowing shifts between food and other components of household budgets), by including an additional elasticity of *expenditure* (TFE$_e$). Such TFEe are occasionally found in the published literature [28–30], but have not (to our knowledge) been used to rescale food purchasing post-PE matrix application. For example, if the food price index (FPI; or average food price) increases by 5% following a saturated fat (SAFA) tax, and the TFE$_e$ is 0.5 (i.e. expected household expenditure on food will increase by 0.5% for each 1.0% increase in the FPI), then the new total expenditure on food will be 2.5% greater than the starting expenditure due to reduced household consumption of other goods such as housing, savings, holidays, etc. We propose first applying the PE matrix conventionally, then scaling all food consumption by a uniform ratio that ensures total food expenditure increases by 2.5%. There is a strong economic rationale for this approach–all food combined in one grouping of household expenditure that will have its own elasticity of demand based on price.

## Methods for TFEe scalar adjustment case studies

### Food and drink taxes and subsidies

Three food tax/subsidy policies were used by way of demonstration:

1. NZ$2 (in 2011 dollars; equivalent to US$1.45 in 2018 dollars) per 100 gram tax on saturated fat, throughout the food system. Such a tax increased the price of butter by 71%, full fat milk by 19% and sausages by 16%.

2. NZ$0.4 (US$0.29) per 100 gram tax on sugar, throughout the food system. Such a tax increased the price of cordials and fruit drinks by 10.8%, tomato sauce by 12.9% and sugar by 82.9%.

3. 20% subsidy on fruit and vegetables.

### Price elasticities

We used a recent, NZ-specific PE matrix published elsewhere [25], and as shown in S1 Table (standard deviations in S2 Table). Briefly, this PE matrix was generated using a Bayesian approach to a linear almost ideal demand system, whereby priors for demand equation coefficients were generated from a previously published New Zealand food PE matrix [4, 5] informing the analysis of food purchasing data generated in a virtual supermarket experiment that randomized participants to differing price sets for foods. The randomized price sets used in this experiment aimed to maximize price variations on foods often suggested as targets for food subsidies or taxes (e.g. sugar tax, F&V subsidy), approximating the degree of price change we use as case studies in this paper.[23, 31] For computational tractability, and theoretical reasons (i.e. food complements and substitutes for cross-PE elasticity estimation are larger between 'like' foods (e.g. poultry and pork) than 'unlike' foods (e.g. poultry and dairy)), the demand equations were first estimated for 11 hierarchical subsets of food groups, then aggregated to one large 23 by 23 food group PE matrix–as is common practice (for example [6]).

**Disaggregation of the 23-by-23 matrix to a 345-by-345 matrix in the simulation modelling.** At the level of 23 food groups, there is still important heterogeneity within food groups in product-level concentrations of sugar and SAFA per 100 grams (for example low and full fat cheeses fall in the same dairy food group). Therefore, we disaggregated foods and their PE to a much larger 345-by-345 food matrix, based on the consumption data in the NZ National Nutrition Survey (2008/09) (acquired directly from the University of Otago's Life in New Zealand Research Group who conducted the survey; personal communication, Blakey, Smith and Parnell, 2014). For example, full-fat and low-fat versions of dairy products should be taxed differently, and it was necessary to allow for shifts in purchasing within dairy products. Whilst external data for finely disaggregated PE matrices was not available, econometric theory posits that as one keeps disaggregating foods into smaller and smaller subgroupings, the own-PE of each food is expected to increase in absolute value terms.[32, 33] For example, the own-PE of all cheese might be -0.6, but high fat cheese separated out might be -0.65. Why? Because, assuming subgroups in each aggregated category are substitutes, changing the price of just high fat cheese means consumers can swap to low fat cheese. A theoretical constraint in food demand system specifies that the sum of budget share weighted own- and cross-PEs for a food item must be a constant, therefore, positive cross-PEs (e.g. between high fat and low fat cheese when we disaggregate cheeses) will lead to an expectation of larger own-PEs (e.g. the own-PE of high fat and the own-PE of low fat cheese).[34] How much does the own-PE strengthen? Unfortunately, that is difficult to

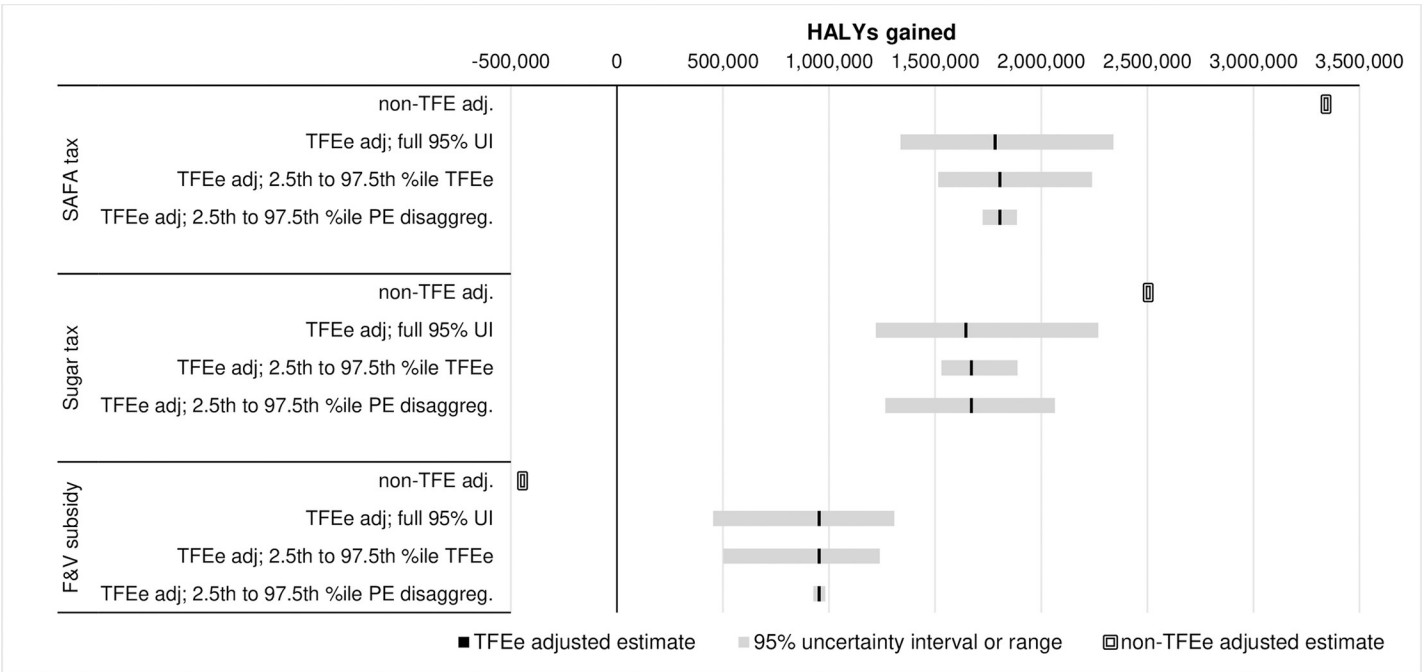

**Fig 1. Central estimate of HALY gained and uncertainty ranges, by policy, for: Non-TFEe adjusted; full probabilistic Monte Carlo simulation for total 95% intervals; univariate sensitivity analysis for 2.5th and 97.5th percentile of TFEe distribution; and univariate sensitivity analysis for 2.5th and 97.5th percentile of PE disaggregation scalar.** The central estimates slightly vary between the 'full' and two sensitivity analyses, as the former is the mean of all Monte Carlo simulations whereas the latter is the central estimate for one simulation using expected (i.e. average) values for all input parameters. Values used to plot this graph are shown in Table 1 and S5 and S6 Tables.

estimate and is genuinely uncertain. Therefore, we first assumed that the own-PE for each of the 23 food groups increases in absolute terms by 2.5% for each additional food sub-group it is disaggregated into, with a relatively wide 1.25% standard deviation (SD) on the normal scale meaning the 95% uncertainty interval traverses 0.05% to 4.95%. (We selected 2.5% as the central value as this would mean a 25% to 50% greater own-PE for each of 10 to 20 disaggregated foods (e.g. the own-PE for types of dairy product considered separately) within one food category (e.g. the own-PE for dairy considered as one group), which seemed plausible given studies that do report both overall and disaggregated own-PEs.) Next, we then ensured the cross-PEs between the newly disaggregated foods satisfied the 'adding up' property with contributions proportional to expenditure (ie, Cournot aggregation).[34] Finally, we report analyses in this paper showing how sensitive results are to this PE disaggregation scalar, by estimating the impact on health adjusted life years (HALYs) gained from using the 2.5th and 97.5th percentile values (i.e. 0.05% to 4.95%; other than the sugar tax, results were reasonably insensitive to its value; Fig 1 and S6 Table).

As an example of how this 2.5% disaggregation scalar works, consider a food group with an aggregated own-PE of -1.0 that is disaggregated to five foods, with proportionate expenditure of 20%, 40%, 20%, 10% and 10% for foods 1 to 5 respectively. The expected own-PE for each of the five foods was -1.125 (i.e. -1–5×0.025), and the cross-PEs for foods 2, 3, 4 and 5 onto 1 were 0.0625 (i.e. 40%/80% × 0.125), 0.03125, 0.015625 and 0.015625 respectively. A similar method was used to disaggregate cross-PE (e.g. the cross-PE of milk onto bread when both food groups were further disaggregated, again ensuring econometric assumptions were met; see elsewhere for details [35]).

**Total food expenditure elasticity (TFEe).**   Estimating the elasticity of expenditure on all foods considered together at the level of changes in the average price of all foods (i.e. TFEe) requires studies of total household expenditure, with consumption items at the level of all foods combined and other groupings of household expenditure (e.g. housing, recreation, education). We are aware of two NZ estimates: Michelini and Chatterjee (1997) and Michelini (1999) [36, 37]. Michelini (1999) is the best with a longer series of data, the use of an almost ideal demands system model, and more disaggregation of food groups. Table 2 of Michelini (1999) reports an own-PE for food combined of -0.168 (standard error 0.1952), which equates to a $TFE_e$ of 0.832 (with the same standard error, which translates to a 95% confidence interval of 0.45 to 1.21). This central value equates to 0.832% increase in total household spending on food for a 1% increase in the FPI. However, the upper confidence limit seems unlikely, as a $TFE_e$ greater than 1 suggests people over-compensate for price increases by spending even more than necessary to maintain the same quantity of food purchased. We also found eight international studies that used multi-stage budgeting models to estimate unconditional and uncompensated food own-PEs, for high-income countries up to June 2017 (keywords: "price elasticities" or "price elasticity" or "demand" and "food" and "multi-stage" or "multi stage"; see Table 2 and adjacent text of [35] for further details). Consistent with theoretical expectation, all estimates were between zero and one. The estimates ranged from 0.46 to 0.90 (except a UK outlier of 0.04), with the average, median and standard deviation across these eight studies being 0.59, 0.66 and 0.29, respectively. For Monte Carlo analyses incorporating input parameter uncertainty, we therefore specified a Beta distribution for the $TFE_e$, parameterized alpha = 6 and beta = 2, which returns mean = 0.75, median = 0.77, mode = 0.83, 2.5[th] percentile = 0.42 and 97.5[th] percentile = 0.96.

## Separate food group expenditure elasticities

When using TFEe, from an econometric perspective we are shifting from conditional to unconditional price elasticities (as we allow total food expenditure to change). This change in total food expenditure it similar to a change in total income, which has a separate impact on food purchasing over and above the own- and cross-PEs. We calculated such expenditure elasticities (EE) at the level of the 23 food groups (S3 Table).

## Estimating post-tax or post-subsidy food quantities

The estimation of post-tax or post-subsidy food quantities is a two-step process. First, we propagate the food price changes through the Marshallian conditional PEs–which should result in no change in total food expenditure (due to the 'conditional' nature of these PEs), but is unlikely to do so for the reasons outlined above and in the Text Box. So we have to rescale all step 1 estimates to ensure this assumption is met. In the second step, we allocate the actual change in total food expenditure out across all foods using the above EEs propagated through the change in total food expenditure (given by the change in food price index and TFEe). However, this is unlikely to generate the 'correct' total post-tax or -subsidy food expenditure, so a further scaling step is required. These two steps are now outlined in detail.

**Step 1: Application of price elasticities and scaling to ensure no change in total food expenditure.** The before-tax or -subsidy total expenditure on food is:

$$X^B = \sum_{i}^{n} q_i^B p_i^B \qquad \text{Eq 1}$$

Where $q_i^B$ is the before tax- or -subsidy quantity and $p_i^B$ is the before-tax or -subsidy price per unit quantity, and i indexes foods 1, 2, 3,. . .,n.

The initial estimate of the percentage change in quantity of each food i due to price changes propagated through price elasticities is:

$$\%\Delta q_{i,PE} = \sum_{j}^{n} (\%\Delta p_j \times \varepsilon_{ij}) \qquad \text{Eq 2}$$

where $\%\Delta p_j$ is the percentage change in price of food j, and $\varepsilon_{ij}$ are the food price elasticities for pairwise combinations of foods (i and j). However, it is not guaranteed that these food quantity changes when multiplied through the after-tax or -subsidy food prices will sum to the same total food expenditure before-tax or -subsidy, which is the assumption of a conditional Marshallian PE. (We need to satisfy this assumption before applying the expenditure elasticities that will 'convert' the calculations from conditional to unconditional.) So, we calculate the implied total food expenditure arising from Eq 2 as:

$$X_{PE} = \sum_{i}^{n} (1 + \%\Delta q_{i,PE}) \, q_i^B p_i^A \qquad \text{Eq 3}$$

where $p_i^A$ is the after-tax or -subsidy price of each food item i. We then scale all first estimates of the percentage change in quantity to ensure no change in total food expenditure:

$$\%\Delta q_{i,PE}^S = \left( \frac{X^B}{X_{PE}} \times (1 + \%\Delta q_{i,PE}) \right) - 1 \qquad \text{Eq 4}$$

And therefore:

$$\Delta q_{i,PE}^S = \%\Delta q_{i,PE}^S \times q_i^B \qquad \text{Eq 5}$$

**Step 2: Application of expenditure elasticities and total food expenditure elasticity (TFEe) to allocate change in total food expenditure across foods.** First, we set the change in total food expenditure as:

$$\Delta X = \%\Delta FPI \times TFE_e \times X^B \qquad \text{Eq 6}$$

Where the percentage change in food price index is:

$$\%\Delta FPI = 100\% \times \left( \frac{\sum_i^n q_i^B p_i^A}{X^B} - 1 \right) \qquad \text{Eq 7}$$

The preliminary estimate of the percentage change in food quantity due to change in total food expenditure is:

$$\%\Delta q_{i,EE} = \%\Delta FPI \times TFE_e \times \eta_i \qquad \text{Eq 8}$$

where $\eta_i$ the food EE of food i. (Note that Eq 8 is utilizing our assumption that total food expenditure after-tax or -subsidy is set by the $TFE_e$ and $\%\Delta FPI$.) Thus, the preliminary estimate of EE generated change in food quantity is:

$$\Delta q_{i,EE} = \%\Delta q_{i,EE} \times (1 + \%\Delta q_{i,PE}^S) \times q_i^B$$

The preliminary estimate of change in total food expenditure is:

$$\Delta X_{EE} = \sum_{i}^{n} \Delta q_{i,EE} \times p_i^A \qquad \text{Eq 9}$$

which is unlikely to equal our target change of $\Delta X$ (Eq 6). We therefore create a scalar:

$$Scalar = \frac{\Delta X}{\Delta X_{EE}}$$ Eq 10

and multiply all preliminary EE-generated changes in q by this scalar to generate 'true' after-tax or -subsidy q:

$$q_i^A = (Scalar \times \Delta q_{i,EE}) + \Delta q_{i,PE}^S + q_i^B$$ Eq 11

that ensures total food expenditure post-tax or -subsidy is:

$$X^A = X^B + \Delta X = \sum_i^n q_i^A p_i^A$$ Eq 12

### Dietary and epidemiological modelling

We estimate the impact of changes in dietary intake and HALYs gained over the remainder of the lives of the New Zealand 2011 population, using a multistate lifetable simulation model. This is a major modelling task. For this paper with objectives of quantifying the impact of TFEe adjustment, and quantifying uncertainty about the same, we only briefly describe this modelling. The conceptual model is shown in S1 Fig, and details are published elsewhere.[35] The starting food consumption was the average (by sex, age, and ethnic group) from the 2008/09 National Nutrition Survey [21], and starting food prices from Nutritrack (a brand-specific packaged food database). Each food was linked to nutrient information using the food composition data from the National Nutrition Survey. By summing across all changes in food intake, several of the outputs presented in this paper were generated: change in total food expenditure, energy intake, and BMI. (The latter BMI change was derived from the change in energy intake, using the method of Hall et al 2011.[38])

The HALYs were estimated using a multi-state lifetable with 14 parallel diet-related diseases, for the entire New Zealand population alive in 2011 modelled over the remainder of their lifetimes. Changes in food, nutrients and physiological measures were combined with relative risks for each of these factors with the diseases (sourced from the Global Burden of Disease study [39]) to generate potential impact fractions (PIFs). These PIFs then altered disease incidence rates (with time lags into the future, e.g. 10 to 30 years (each limit with probabilistic uncertainty) for cancers), which then altered disease prevalence rates and mortality rates, all captured in the main lifetable as incremental changes in HALYs.

Monte Carlo simulation was used to estimate uncertainty in the HALY outputs, by drawing randomly from probability distributions about all input parameters (some given above; others elsewhere [35]; 2000 iterations). In addition to generating 95% uncertainty intervals about the HALYs, we also explore the specific impact of uncertainty in the TFEe and PE disaggregation scalar by reporting the HALY values for the 2.5th and 97.5th percentiles of these two input parameters (i.e. univariate sensitivity analyses).

### Results

Table 1 shows the impacts of the three tax and subsidy scenarios onto changes in expenditure, grams of food per day, energy intake, BMI and HALYs gained–before (i.e. conventional analyses) and after TFEe adjustment. Consider first the changes in expenditure in the first column. The SAFA tax resulted in an increase of the total price index (i.e. price of all food together) of 3.91%, yet a conventional application of PEs suggests that the consumer will not compensate at all for this and instead decrease total food expenditure by 1.92% (Table 1). Put another way,

**Table 1. Model outputs (grams of food/day, expenditure, energy, BMI and HALYs gained) for saturated fat and sugar taxes, and fruit and vegetable subsidy, for the preferred TFEe adjustment and conventional (no TFEe adjustment) analyses.**

| | Expendi-ture # | All food (g/day) | Energy (kJ) | BMI | Fruit (g/day) | Vege (g/day) | Salt (g/day) | PUFA (g/day) | SSBs (mls/day) | Sugar (g/day) | HALYs † | 95% UI HALYs ‡ |
|---|---|---|---|---|---|---|---|---|---|---|---|---|
| **Business as usual (BAU)** | 16.09 | 3016 | 8,536 | 27.51 | 149.44 | 149.68 | 3.43 | 0.050 | 102.58 | 108.92 | 173,012,000 | |
| **Changes compared to BAU** | | | | | | | | | | | | |
| ***Saturated fat tax of $2 per 100g** (causing a 3.91% increase in the FPI)* | | | | | | | | | | | | |
| Conventional model–no TFEe adjustment | -1.92% | -68 | -740 | -1.30 | -0.58 | -0.83 | -0.19 | -0.001 | 0.40 | -5.21 | 3,343,000 | |
| TFEe adjustment | 2.93% | -14 | -348 | -0.61 | 5.75 | 6.20 | -0.07 | -0.001 | 4.65 | 0.18 | 1,805,000 | (1,337,000 to 2,340,000) |
| ***Sugar tax of $0.4/100 grams per 100g** (causing a 1.88% increase in the FPI)* | | | | | | | | | | | | |
| Conventional model–no TFEe adjustment | -1.04% | -45 | -522 | -0.91 | -0.05 | 0.00 | -0.04 | 0.002 | -22.50 | -17.53 | 2,504,000 | |
| TFEe adjustment | 1.41% | -16 | -321 | -0.56 | 3.20 | 3.58 | 0.02 | 0.002 | -20.56 | -20.81 | 1,671,000 | (1,220,000 to 2,269,000) |
| ***Fruit and vegetable subsidy of 20%** (causing a 3.27% decrease in the FPI)* | | | | | | | | | | | | |
| Conventional model–no TFEe adjustment | 0.72% | 78 | 218 | 0.39 | 28.72 | 53.97 | 0.04 | 0.000 | -0.46 | 5.49 | 415,000 | |
| TFEe adjustment | -2.45% | 45 | -56 | -0.10 | 24.22 | 48.62 | -0.05 | 0.000 | -2.88 | 1.84 | 953,000 | (453,000 to 1,308,000) |

† 0% discount rate; HALYs at 3% annual discount rate are shown in S4 Table. Values are 'expected values' using central estimates for all input parameters (i.e. not from Monte Carlo simulation).

‡ Uncertainty intervals for 2000 simulations (for TFEe adjusted results only) drawing the 2.5th and 97.5th percentiles.

# $ for BAU. % change for changes compared to BAU.

this is a 'revealed' TFEe of -0.49 (i.e. -1.92/3.91) meaning that for every 1% increase in the overall price of food the consumer will actually reduce expenditure by 0.49%. Our central estimate of the TFEe is 0.75 (i.e. consumers will increase food expenditure by 0.75% for every 1% increase in the food price index), which when used to rescale all expenditure results in a post-TFEe adjusted increase of 2.93% in food expenditure. (We consider uncertainty in this 0.75 estimate below.) Looking down the rest of first column of Table 1 we see that in all instances the conventional PE application shifts the total food expenditure in the opposite direction to theoretical expectation, namely 'revealed' TFEs of -0.56 and -0.22 for the sugar tax and F&V subsidy.

Rescaling all food purchasing by a common amount so that the overall change in total food expenditure is 75% of the change in food price index, the TFEe adjustments result in: more muted changes in grams of food but no change of direction; lesser reductions in energy intake and BMI for SAFA and sugar tax, and a reversal to a modest reduction in energy intake and BMI for the F&V subsidy.

Regarding the flow on impact to HALYs gained (Table 1 and Fig 1), TFEe adjustment reduced the HALY gains by 46% for the SAFA tax and by 33% for the sugar tax, and reversed an apparent health loss to a health gain for the F&V subsidy. TFEe adjusted HALY gains (in 1000's) were: 1,805 (95% uncertainty interval 1,337 to 2,340) for the SAFA tax; 1,671 (1,220 to 2,269) for the sugar tax; and 953 (435 to 1,308) for the F&V subsidy. It is important, however,

to put these HALY <u>gains</u> in context of the 173 million HALYs under <u>business as usual</u> over the remainder of the population's lifespan. Accordingly, reconsider the SAFA tax: the conventional analysis was suggesting a 1.93% increase in HALYs, and the TFEe adjustment a 1.05% increase in HALYs–or a 0.88% point difference. And for the F&V subsidy: the conventional analysis was suggesting a 0.24% decrease in HALYs, and the TFEe adjustment a 0.55% increase in HALYs–or a 0.79% point difference. That is, we are trying to pick up relatively small magnitude effect sizes, and a small absolute error or difference appears as a large relative error or difference.

Fig 1 also shows the 95% uncertainty intervals for the TFEe adjusted analysis (i.e. for Monte Carlo simulation where all input parameters are sampled from their uncertainty distributions), plus the range of HALY gain values for the 2.5$^{th}$ to 97.5$^{th}$ percentile values of the probability distributions for the TFEe (2.5$^{th}$ percentile = 0.42 and 97.5$^{th}$ percentile = 0.96), and the PE disaggregation scalar (0.05% and 4.95%). The uncertainty in the TFEe accounts for much of the total 95% uncertainty for the SAFA tax and the F&V subsidy. The uncertainty in the PE disaggregation parameter is less influential, except for the sugar tax where it drives much of the total uncertainty (including more uncertainty than that for the 95% range of TFEe values).

## Discussion

Our modelling suggests that conventional application of PE matrices can produce implausible absolute changes in food nutrient intake. Based on the rationale that PE matrix induced changes in relative food intake are valid for food tax and subsidy policies simulated with an appropriate PE matrix (e.g. that a saturated fat tax reduces fatty food purchasing and intake relative to F&V), we present the TFEe scalar as a theoretically plausible solution to scale or constrain total absolute expenditure and therefore total food and nutrient intake. Economic theory suggests that change in total food expenditure will follow change in the total FPI according to an expenditure elasticity, the TFEe. Our empirical simulations using a TFEe adjustment have face validity. For example, the magnitude of HALY gains is maximal for sugar and SAFA taxes that impact many foods.

There are limitations to our proposed method and modelling. First, there may not be a ready source TFEe for a given context. However, there are strong theoretical bounds for the TFEe: it is unlikely to be greater than 1.0 –as this would imply that consumers increase food expenditure by a greater percentage than the percentage change in total food price index; and it is unlikely to be less than 0 –as this would imply that consumers do not increase food expenditure at all in response to a price increase. Thus, it is likely bounded between 0 and 1, and with plausible uncertainty intervals about its value (as we argue we included in this study) the true value will be covered. Put another way, to not use TFEe scaling will often equate to assuming that it is less than 0 or greater than 1.0 (e.g. as implied by some of our unscaled analyses). Second, our study is just for NZ PEs applied to NZ; attempts at replication in other contexts are justified. Third, we assume perfect competition in that the pre-tax market price does not vary once taxes (or subsidies) are imposed, that tax pass-through is 100%, and that the price elasticities we use (with TFEe adjustment) are reasonable proxies for long-run responses to price changes. Fourth, more dietary risk factors could be included than shown in our conceptual model to estimate HALYs (S1 Fig). For example, a recent global burden of disease study includes 15 dietary risk factors with updated relative risks.[40] Including more risk factors will probably just increase the absolute magnitude of HALY changes, but not the alter the pattern of findings–unless a specific tax was placed on a food item not currently in our model (e.g. red meat). Fifth, our study is a modelling study. There is a strong need for more real-world evaluations of food taxes and subsidies. One way forward is to use natural experiment analyses that

carefully evaluate the impact of actual policies, for example Smed et al (2016) used econometric fixed effects modelling of consumer purchasing data to evaluate the Danish saturated fat tax. [16] They found that the tax did reduce saturated fat purchasing, and that it also had mixed spillover effects to increase vegetable consumption but also increase salt intake. Interestingly in light of our current study, they found it necessary to constrain or rescale their outputs to an assumption of no change in energy intake when modelling deaths averted or delayed, as sensitivity analyses found the net health impact to be very sensitive to any change in energy intake. Put another way, even though Smed et al undertook as rigorous as possible natural experiment analyses, they still confronted a need for theoretical constraint or scaling of the modelling of health impacts–demonstrating both the challenges in estimating net heath impacts, and also the need for studies such as ours probing deeper the issue and possible solutions.

It is useful to compare our method with some previously published simulation studies using PE matrices (Table 2). Approaches used that may prevent undue violation of assumptions inherent in the calculation of PE matrices include ensuring the total changes in food price for any tax or subsidy policy is less than 1% [9] and policies that use F&V subsidies to offset food tax revenue.[8] (Expressed conversely, any food and tax subsidy policy that sees a sizeable change in the total food price index likely requires some form of constraint–which we propose may be the TFEe.) Briggs et al (2015)[6]–whilst only assessing the impact of taxes on sugary drinks (a small fraction of all food)–used hierarchical demand systems (e.g. solving separate drinks as one system separately from drinks combined with other food groups). This hierarchical approach means that cross-PEs between what we think are disparate food groupings (e.g. meat versus breakfast cereals) are actually constrained to smaller absolute values, as substitutes and complements are assumed to mainly occur *within* the separate food demand systems.

**Table 2. Characteristics of selected previous food tax and subsidy modelling papers.**

| | | Price elasticity matrix | | | | |
|---|---|---|---|---|---|---|
| Author | Interventions and setting | Number of food groups | Derivation of PE matrix | Cross-PE used? | Constraint or rescaling after PE application? | Health gain findings |
| Blakely et al (current study) | NZ. SAFA and sugar tax, F&V subsidy. | 340 (disaggregated from 23) | Bayesian LAIDs model, 12 hierarchical demand systems. Marshallian conditional PEs. | Yes | Yes; using TFEe | Substantial HALY gains: SAFA tax ≈ sugar tax > F&V subsidy. |
| Briggs et al (2013) [6] | UK. 20% sugar sweetened drink tax. | 12 drinks categories and 5 food categories. | Bayesian AIDs model, 5 hierarchical demand systems. Unconditional within each demand system; conditional across demand systems. | Yes | No | 20% SSB tax would result in 1.3% reduction in obesity rates. |
| Cobiac et al (2017) [9] | Australia. Separate and combined policies such that all policies had <1% impact on total food expenditure. Salt, sugar, saturated fat and SSB taxes: F&V subsidy. | 24 | NZ PE matrix as used in Ni Mhurchu et al (2015) [4]. UK PE matrix for sensitivity analysis. | Yes, with suppression of statistically non-significant cross-PE. | No | Combined taxes and F&V subsidy > sugar tax > salt tax ≈ SAFA tax. F&V subsidy alone led to health loss. Sensitive to PE matrix used. |
| Ni Mhurchu et al (2015) [4] | NZ. Sodium and sugar tax. F&V subsidy. Tax on foods contributing to greenhouse gases. | 24 | Household economic survey data, with prices from food price index | Yes, with theoretical suppression of non-important cross-PE. | No | Sodium tax > sugar tax > F&V subsidy in terms of deaths prevented or postponed. |

AIDS = almost ideal demand system. LAIDS = linear AIDS. HALY = health adjusted life year.

Unconditional means that the a change in expenditure was allowed in the assumptions for calculating PE.

Our study here offers three advances. First, the price elasticities we draw on use a Bayesian method blending priors with experimental data, and a hierarchical of demand systems that should (consistent with theory) mean cross-PEs between disparate foods are appropriately kept small in absolute terms (similar to Briggs et al above [6]). Second, we present a method for disaggregating a PE matrix to a large (in this case 340 food groups) matrix to allow separate consideration of foods with differing nutrient levels (e.g. sugar content) that consequently attract differing price changes. Third, and the main purpose of this paper, we propose using an additional constraint–the TFEe. We propose that there is good a priori reason to use this constraint, and particularly so: a) if the policies we are assessing generate sizeable changes (e.g. greater than 1%) in the FPI; and/or 2) the modelled total food expenditure following conventional application of the PE matrix is 'counter-intuitive' economically, for example with total expenditure decreasing post tax or total expenditure increasing post-subsidy.

In conclusion, we are proposing new methods that we think can improve simulation of food tax and subsidy impacts. We encourage other researchers to scrutinize and critique our proposals, and we strongly recommend future research to compare estimates from such modelling with real-world natural experiments.

## Supporting information

**S1 Fig. Conceptual diagram of the model.**
(DOCX)

**S1 Table. Marshallian cross- and own-PEs matrix using a Bayesian Linear Almost Ideal Demand System (LAIDS) approach.**
(DOCX)

**S2 Table. Standard deviations about the Marshallian cross- and own-PEs shown in S1 Table.**
(DOCX)

**S3 Table. Food group level (n = 23) expenditure elasticities applied after conventional application of the PE matrix, and before the final TFEe scaling.**
(DOCX)

**S4 Table. HALYs gained at 3% discount rate for saturated fat and sugar taxes, and fruit and vegetable subsidy, for the preferred TFEe adjustment and conventional (no TFEe adjustment) analyses.**
(DOCX)

**S5 Table. Univariate sensitivity analyses about low and high (2.5th and 97.5th percentile) values of TFEe, using the "TFEe adjustment" models.**
(DOCX)

**S6 Table. Univariate sensitivity analyses about low and high (2.5th and 97.5th percentile) values of PE disaggregation scalar.**
(DOCX)

## Acknowledgments

Nick Wilson, Wilma Waterlander, Liana Jacobi, Andres Ramirez Hassan, Dietary intake data used in this modeling was supplied by Otago University 'Life in New Zealand' staff (personal communication, Blakey, Smith and Parnell, 2014).

## Author Contributions

**Conceptualization:** Tony Blakely, Boyd Swinburn, Peter Scarborough, Christine Cleghorn.

**Data curation:** Christine Cleghorn.

**Formal analysis:** Tony Blakely, Nhung Nghiem, Murat Genc, Anja Mizdrak, Linda Cobiac, Christine Cleghorn.

**Funding acquisition:** Tony Blakely.

**Methodology:** Tony Blakely, Christine Cleghorn.

**Project administration:** Christine Cleghorn.

**Supervision:** Christine Cleghorn.

**Writing – original draft:** Tony Blakely.

**Writing – review & editing:** Tony Blakely, Nhung Nghiem, Murat Genc, Anja Mizdrak, Linda Cobiac, Cliona Ni Mhurchu, Boyd Swinburn, Peter Scarborough, Christine Cleghorn.

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
