## [Decision Letter · Decision Letter 0]

11 Dec 2019

PONE-D-19-27149

Modelling the health impact of food taxes and subsidies with price elasticities: the case for additional scaling of food consumption using the total food expenditure elasticity

PLOS ONE

Dear Prof. Blakely,

Thank you for submitting your manuscript to PLOS ONE. After careful consideration, we feel that it has merit but does not fully meet PLOS ONE’s publication criteria as it currently stands. Therefore, we invite you to submit a revised version of the manuscript that addresses the points raised during the review process.

We would appreciate receiving your revised manuscript by Jan 25 2020 11:59PM. To enhance the reproducibility of your results, we recommend that if applicable you deposit your laboratory protocols in protocols.io, where a protocol can be assigned its own identifier (DOI) such that it can be cited independently in the future. For instructions see: http://journals.plos.org/plosone/s/submission-guidelines#loc-laboratory-protocols

We look forward to receiving your revised manuscript.

Kind regards,

Chuanwang Sun

Academic Editor

PLOS ONE

Journal Requirements:

Please ensure that your manuscript meets PLOS ONE's style requirements, including those for file naming. The PLOS ONE style templates can be found at http://www.plosone.org/attachments/PLOSOne_formatting_sample_main_body.pdf and http://www.plosone.org/attachments/PLOSOne_formatting_sample_title_authors_affiliations.pdf Please include captions for your Supporting Information files at the end of your manuscript, and update any in-text citations to match accordingly. Please see our Supporting Information guidelines for more information: http://journals.plos.org/plosone/s/supporting-information

Reviewers' comments:

Reviewer's Responses to Questions

5. Review Comments to the Author

Reviewer #1: The authors adopted price elasticity (PE) matrices to study food taxes. It ‘s an interesting topic and it’s useful for developing countries and areas. The authors may contribute to build the models, but the analysis in the paper is built on literatures and lack of innovation. Furthermore, the conclusions need to be more clear and concrete to show the final findings and economic meanings of the research.

Reviewer #2: This paper uses previously estimated price elasticities for food products to estimate the effect on diets from a taxes on saturated fat and on sugar and a subsidy on fruits and vegetables. The paper goes on to estimate the effect of the estimated change in diets on health, in particular adjusted health years. The innovation in the paper is to adjust the price elasticities to account for the change in total expenditures on food from the change in aggregate food prices, which the authors refer to as the total food elasticity adjustment, TFEe adjustment. The authors find that the adjustment has a significant effect on the price elasticities they use to calculate changes in health.

While the author discuss the TFEe adjustment, I would have liked to have seen more detail regarding how the calculation is made. (On page 15, the authors claim that their approach is a “theoretically plausible solution.” But they don’t demonstrate that point.) If the adjustment or calculation were made using estimated share rather than price elasticities, the calculation would go something like this: Let sit be the share of total food expenditures Et in year t spent on product i. The available estimates of s assume that E is fixed. Thus is it necessary to account for the effect of price change on both s and E in order to measure the effect of the price change on total quantity q consumed of product i. The change in q can be expressed as

Δq=s_(t+1)*E_(t+1)-s_t*E_t=Δs*E_t+s_t*ΔE+Δs*ΔE

An equivalent expression using elasticities would have been helpful.

There is an implicit assumption that the pre-tax market price does not change in response to the taxes and subsidies. It is reasonable to assume perfect competition and constant marginal cost in food production NZ?

Another implicit assumption is that the estimated price elasticities apply in the long run. If I understand how the price elasticities were estimated, I believe that they reflect short run responses to price changes. But, it is expected that over time, consumers will modify their response to the initial price change. This is important since the price elasticities are used to estimate changes in diet over an individual’s lifetime.

If expenditures on food increase as a result of the tax/subsidy policy, might the change in non-food spending result in health effects? For example, might there be less spending on exercise classes or health care, or less driving but more walking?

It would be desirable to compare the change in prices used in the research that generated the price elasticities to the implicit change in prices from the tax/subsidy policy. In particular, are the authors trying to estimate effects that are well outside the range of prices used for the price elasticity estimation?

Regarding the writing, in general, I found it difficult to understand what the authors were attempting to do and then how they did it. The paper could have been made much clearer.

The paper uses estimates that were generated in other papers, for example the price elasticities. I found it difficult to follow the explanation of the price elasticity estimates. It was technical and short compared to the original papers, making it hard to obtain an intuitive understanding of the procedure. I think it might have been desirable if the discussion of how those results were derived was separated from the rest of the discussion.

There are many typos and places where the writing could be made clearer and the presentation improved. Here are some examples, but this is not a comprehensive list:

P3. HALY is defined in the abstract. It should be defined the first time it is used in the body of the paper.

P6, 1st paragraph. The explanation of price elasticity is very basic and could be dropped or relegated to a footnote. On the other hand the explanation for the need to scale the estimated elasticities is not well articulated.

P6, 2nd paragraph. The following is unclear, “…on both price and demand or purchasing with sufficient variation in price…” as is “…starting proportionate consumption…”

P6, 2nd paragraph. The sentence beginning with “Indeed, the most common…” could be improved by more clearly noting that it is in reference to studies that focus just on food consumption. I first thought it meant general studies of price elasticities that assume no change in food expenditures, which didn’t make any sense.

P6, 3rd paragraph. Consider the sentence, “Is applying PEs derived from one setting into another acceptable?” I found this misleading. At first I thought the task was that the paper was taking price elasticity estimated from one data source, and using them to estimate changes using a different data set. But that is not exactly what is being done. The paper does not apply them in another “setting”. I struggled with trying to understand what the authors were doing, and found the writing confusing rather than enlightening.

P7-8. Given that the objective is to estimate the effect on consumption of food items from a price change, the suggestion that one would do assuming that energy intake or grams of food purchased is constant, makes no intuitive sense. I don’t see the point of discussing those two options.

P10, 2nd line. “more important” is not what you mean, but rather “larger”.

P13, Results. Information on the consumption of saturated fats and sugar would have been helpful in understanding how the tax results in a price increase of 3.91% for all food.

P22, Text Box. The text says a 20% subsidy, while the spreadsheets refer to a 10% subsidy.

---

## [Decision Letter · Decision Letter 1]

3 Mar 2020

Modelling the health impact of food taxes and subsidies with price elasticities: the case for additional scaling of food consumption using the total food expenditure elasticity

PONE-D-19-27149R1

Dear Dr. Blakely,

We are pleased to inform you that your manuscript has been judged scientifically suitable for publication and will be formally accepted for publication once it complies with all outstanding technical requirements.

With kind regards,

Chuanwang Sun

Academic Editor

PLOS ONE

Additional Editor Comments (optional):

Reviewers' comments:

Reviewer #2: The authors have address all of my comments on the first version of the paper. I have no further comments.

---

## [Editor Report · Acceptance letter]

10 Mar 2020

PONE-D-19-27149R1 

Modelling the health impact of food taxes and subsidies with price elasticities: the case for additional scaling of food consumption using the total food expenditure elasticity 

Dear Dr. Blakely:

I am pleased to inform you that your manuscript has been deemed suitable for publication in PLOS ONE. Congratulations! Your manuscript is now with our production department. 

With kind regards,

on behalf of

Dr. Chuanwang Sun 

Academic Editor

PLOS ONE